# Injectable Platelet-Rich Fibrin as a Drug Carrier Increases the Antibacterial Susceptibility of Antibiotic—Clindamycin Phosphate

**DOI:** 10.3390/ijms23137407

**Published:** 2022-07-03

**Authors:** Karina Egle, Ingus Skadins, Andra Grava, Lana Micko, Viktors Dubniks, Ilze Salma, Arita Dubnika

**Affiliations:** 1Rudolfs Cimdins Riga Biomaterials Innovations and Development Centre, Institute of General Chemical Engineering, Faculty of Materials Science and Applied Chemistry, Riga Technical University, LV-1007 Riga, Latvia; karina.egle@rtu.lv (K.E.); andra.grava@rtu.lv (A.G.); viktors.dubniks@rtu.lv (V.D.); 2Baltic Biomaterials Centre of Excellence, Headquarters at Riga Technical University, LV-1048 Riga, Latvia; ingus.skadins@rsu.lv (I.S.); lana.micko@gmail.com (L.M.); ilze.salma@rsu.lv (I.S.); 3Department of Biology and Microbiology, Riga Stradins University, LV-1007 Riga, Latvia; 4Institute of Stomatology, Riga Stradins University, LV-1007 Riga, Latvia; 5Department of Oral and Maxillofacial Surgery, Riga Stradins University, LV-1007 Riga, Latvia

**Keywords:** platelet-rich fibrin, antibacterial properties, antibiotic resistance, drug release, CLP

## Abstract

The aim of this study was to investigate the change in clindamycin phosphate antibacterial properties against Gram-positive bacteria using the platelet-rich fibrin as a carrier matrix, and evaluate the changes in the antibiotic within the matrix. The antibacterial properties of CLP and its combination with PRF were tested in a microdilution test against reference cultures and clinical isolates of *Staphylococcus aureus* (*S. aureus)* or *Staphylococcus epidermidis* (*S. epidermidis)*. Fourier-transform infrared spectroscopy (FTIR) and scanning electron microscope (SEM) analysis was done to evaluate the changes in the PRF_CLP matrix. Release kinetics of CLP was defined with ultra-performance liquid chromatography (UPLC). According to FTIR data, the use of PRF as a carrier for CLP ensured the structural changes in the CLP toward a more active form of clindamycin. A significant decrease in minimal bactericidal concentration values (from 1000 µg/mL to 62 µg/mL) against reference cultures and clinical isolates of *S. aureus* and *S. epidermidis* was observed for the CLP and PRF samples if compared to pure CLP solution. In vitro cell viability tests showed that PRF and PRF with CLP have higher cell viability than 70% after 24 h and 48 h time points. This article indicates that CLP in combination with PRF showed higher antibacterial activity against *S. aureus* and *S. epidermidis* compared to pure CLP solution. This modified PRF could be used as a novel method to increase drug delivery and efficacy, and to reduce the risk of postoperative infection.

## 1. Introduction

Platelet-rich fibrin (PRF) is an autogenous material derived from human blood and is widely used to promote wound healing and tissue regeneration [1]. The leukocytes in the PRF promote wound healing and PRF contains growth factors that are released over time [2]. In several applications, such as oral and maxillofacial surgery, plastic surgery, cardiac surgery and dentistry, there is a great interest in PRF antimicrobial activity. Until now, most clinical studies have been conducted in dentistry and oral and maxillofacial surgery. Platelet concentrates are used in maxillary sinus floor augmentation, as the filling of teeth extraction sockets, in dental implant surgery, in regenerative endodontic treatment, in peri-implantitis and periodontitis treatment.

During the last decade, the antimicrobial properties of PRF have been described in various studies and different testing methods and bacteria have been used. Not only is injectable platelet-rich fibrin (I-PRF) anti-microbial, but anti-biofilm activity against human oral abscess pathogens has also been described. It was found that I-PRF decreases biofilm production at the minimal inhibitory concentration (MIC) and no biofilm production at the minimal bactericidal concentration (MBC) [3]. Using the disk diffusion method, I-PRF showed notable zones of inhibition, which varied depending on different bacterial species [4]. I-PRF also shows the superiority of antimicrobials against bacteria from the supragingival plate over PRF and PRP [4].

Infections are one of the most common postoperative risks caused by pathogenic and opportunistic bacteria [5,6]. *S. aureus* and *S. epidermidis* are Gram-positive opportunistic bacteria, which are present in the normal human microbiome. With the ability to produce biofilms, these bacteria can evade the host immune system and can cause various local and systemic infections, such as bacteremia, skin and soft tissue infections, osteomyelitis, and implant and device-related infections [7,8,9]. A lot of these infections can be prevented with antibiotics, especially those where the portals of entry for the bacteria are wounds due to surgery in the hospital environment. For treatment of community-acquired, methicillin-resistant and methicillin-susceptible *S. aureus* infections, clindamycin has been recommended for many years, and it also can increase the susceptibility of methicillin-resistant *S. aureus* clinical isolates [10,11]. The virulence of clinical isolates and their virulence factors, such as surface proteins, determine their ability to cause disease and the severity of the disease [12]. Nowadays, the demand for clindamycin as a medicine is increasing in oral and maxillofacial surgery (including for the prevention and treatment of osteonecrosis of the jaw [13]). It is widely considered as an alternative for patients with an allergic reaction to penicillin [14].

Clindamycin phosphate (CLP) is a prodrug of clindamycin that has no antibacterial activity [15]. As mentioned in the literature, prodrugs can offer many advantages over the parent drug (in our case clindamycin). These benefits include increased solubility, improved stability, reduced side effects, improved bioavailability and better selectivity [16]. CLP can be converted to clindamycin by in vitro hydrolysis of phosphatase esters [15,17]. Rapid in vivo hydrolysis also converts the CLP compound to the antibacterial clindamycin. Hydrolysis of the phosphatase ester is a relatively difficult mechanism (Figure 1). It has been reported that after hydrolysis using alkaline phosphatase, clindamycin phosphate is determined as clindamycin. Following topical, as well as upon intravaginal administration, clindamycin phosphate is slowly hydrolyzed to clindamycin due to limited hydrolysis of the prodrug by the phosphatase enzymes on the surface and within the skin. This prevents the incidence of antibiotic-induced GI side effects [18].

Clindamycin is known to be obtained by the chemical modification of lincomycin, so the potential impurities are analogs of lincomycin [19]. It is reported that the conversion of clindamycin phosphate to clindamycin in the blood is significantly lower than with oral administration of clindamycin hydrochloride [20,21]. CLP is absorbed as an inactive ester for parenteral use and is rapidly hydrolyzed to the active base in the blood.

CLP has not been widely studied in terms of antibacterial properties. Until now there are only a few studies on CLP’s individual antibacterial properties, which show that the MIC of CLP (*w*/*v* %) on *Staphylococcus aureus* is 0.02 ± 0.005% [22]. Nevertheless, it has a broad range of applications in biomaterials: for example, it is used in eye implants [23], periodontal films [24], and particles [25].

The aim of this study was to investigate the change in CLP antibacterial properties against reference culture and clinical isolates of *S. aureus* and *S. epidermidis* using platelet-rich fibrin as a carrier matrix, and evaluate the CLP structural changes, release kinetics and in vitro cytotoxicity within the PRF matrix.

## 2. Results

### 2.1. Structural Changes in PRF and PRF_CLP Samples at 37 °C

The FTIR spectrum of PRF and PRF_CLP is shown in Figure 2. The FTIR spectra of the samples in Figure 2A,B show the absorption peaks indicating the fibrin phase: peak at 1641 cm^−1^—amide I (C=O), maximum at 1535 cm^−1^—amide II (N-H) and amide III (C-N) (decreases at 1310 cm^−1^ and increases at 1236 cm^−1^) (Figure 2B). Characteristic changes in the FTIR spectra are due to rearrangements in the secondary structure of the protein. According to other studies [26,27], the absorption of different proteins at higher wavelengths (1633–1645 cm^−1^, 1531–1539 cm^−1^ and 1240 cm^−1^) occurs mainly due to α-helical structures, whereas the lower wavenumbers (1651 cm^−1^, 1539 cm^−1^) are mostly characteristic of β-structures [28]. Thus, the α-structure is more pronounced in the studied sample. A pronounced absorption maximum at 3281 cm^−1^ indicates the presence of an OH group in the fibrin structure.

CLP and clindamycin have a similar molecular structure, except for the phosphate group. The absorption maximum, specific for both CLP and clindamycin, was observed in PRF_CLP samples incubated for 1, 3, and 7 days at 37 °C. The main structural components of clindamycin molecules are characterized by the vibrations of the pyrrole and saccharide rings, which form skeletal vibrations between 1600 and 600 cm^−1^. The band group indicated in this region is mainly related to C double bond tensile vibrations. Large changes are observed at about 1047 cm^−1^, which corresponds to the C-C stretching of the pyrrolidine group. It can be seen that as the incubation time of the PRF_CLP samples increases (from 1 to 7 days), the intensity of the C-C bond also increases. The tensile vibrations of the C-O groups bound to the saccharide ring are observed at 1157 cm^−1^ [23]. The band at 640 cm^−1^ corresponds to the tensile vibrations of the C-Cl groups. In the spectra of the PRF_CLP samples, a wide band with a maximum of 3350 cm^−1^ can also be observed, which corresponds to the vibrations of the O-H groups of aromatic alcohols [29].

It is also observed that the intensity of the phosphate group (PO_4_^3−^ at 531 cm^−1^) from CLP increases in the PRF_CLP sample after 7 days of incubation compared to the PRF_CLP sample after 1 day of incubation (Figure 2B). This may be due to the formation of other CLP degradation products or potential contaminants except clindamycin. Wang et al. [30] state that in addition to clindamycin, two other substances are formed: lincomycin-2-phosphate and clindamycin B-2-phosphate. Brown [19], on the other hand, mentioned the formation of three substances—clindamycin 3-phosphate, clindamycin 4-phosphate and clindamycin 2-phosphate. It can be concluded that the possible degradation products increased the intensity of the PO_4_^3−^ absorption peak with increasing degradation time.

It has also been observed that the bands characteristic of clindamycin and CLP at 1673 cm^−1^ (NH-C=O) and at 1568 cm^−1^ (C-C) shift to the right in the presence of PRF. The spectra of the PRF_CLP samples show an increase in the absorption peaks of the above bands, which may have been influenced by the interaction of PRF with CLP. The development of additional intensity at 1080 cm^−1^ C-O cyclic ester galactose sugar elongation [31] was also observed for the PRF_CLP sample after 7 days of incubation. Looking at the CLP and clindamycin spectra, this absorption peak is most indicative of clindamycin. Based on the literature [15,17], CLP induces hydrolysis in the presence of blood and converts to clindamycin. It is possible that this hydrolysis and partial conversion to clindamycin is observed in the spectra of PRF_CLP samples.

SEM images of the PRF matrices with and without CLP after incubation and lyophilization are shown in Figure 3. Examining PRF samples with SEM, it can be seen that their surface morphology is irregular with a porous microstructure. There are no visible differences in the structure of the PRF samples depending on the incubation time (1, 3 and 7 days at 37 °C). For PRF_CLP samples after 1-day incubation, crystalline structure formations can be seen on the surface of the sample (marked in the images with a red line), these are also observed after 3 and 7 days.

Comparing the PRF and PRF_CLP samples, it can be seen that the addition of CLP did not significantly affect the structure of the PRF after 1 and 3 days of incubation. In turn, after 7 days of incubation, small network formations are observed on the surface of the PRF_CLP sample. This could be related to the degradation of CLP, thus changing the structure of the PRF.

According to the SEM-EDX data, the crystalline structures present in the PRF_CLP sample contain a large amount of NaCl. PRF contains Na ions and according to Pradid et al., the Cl peaks indicate the presence of clindamycin phosphate [32].

### 2.2. Drug Release Kinetics

CLP release from PRF matrices was determined by incubating PRF matrices for 0.25, 0.5, 1, 2, 4, 6, 17 and 24 h (Figure 4).

Burst release of CLP was observed for all PRF_CLP samples in the first incubation hour, when 80% of the encapsulated CLP was released. Based on the obtained data, it is possible to provide local antibacterial activity in a certain place in the first hours, thus reducing the risk of infection during the postoperative period. For long-term treatment, drug delivery systems should be used to prevent burst release during the first hours. ANOVA tests show that at *p* < 0.05, there is no significant difference between drug release from different donor samples.

### 2.3. Effect of PRF_CLP on Antibacterial Properties

Four Gram-positive bacterial cultures and three donors were used to evaluate and compare the antibacterial properties between CLP, PRF and PRF_CLP samples (Figure 5 and Figure 6).

As shown in Figure 5, the MIC and MBC values for pure CLP solution were first determined to test the ability of the substance to provide an antibacterial effect against selected bacterial cultures. The data showed that against *S. aureus* (ATCC 25923), *S. epidermidis* (ATCC 12228) and *S. epidermidis* (clinical isolate), higher CLP concentrations (1000 µg/mL) were required if compared to *S. aureus* (clinical isolate)—500 µg/mL. In general, high CLP concentrations (1000 µg/mL) are required for maximal effect.

Obtained results showed that negative control (PRF_CLP_broth solution) has an increased level of absorption in the higher concentrations, due to the autologous PRF sample. This is because the PRF has a color that comes from the blood sample. According to the obtained results for each donor’s antibacterial properties, MIC and MBC levels for PRF_CLP samples depend on the donor and the bacteria strain. In general, we observed that the incorporation of CLP within the PRF leads to lower MIC and MBC values for all donors and all bacteria strains (Figure 6).

The mean donor MIC values for PRF_CLP samples ranged from 52.1 to 62.5 μg/mL, which are lower than the MIC values for pure CLP samples (ranging from 125 to 250 μg/mL). In turn, the mean MBC values range from 62.5 to 145.8 µg/mL, while for pure CLP samples they are at 500—1000 µg/mL. Differences in MIC and MBC values are affected by the bacteria selected for testing (Appendix A).

As shown in Figure 6, there is a difference in the MIC value of the donor 1 PRF_CLP samples against *S. aureus* (ATCC 25923). It is lower (31.25 µg/mL) than against three other bacterial cultures (62.5 µg/mL). For donor 2 PRF_CLP samples, lower MIC (31.25 µg/mL) and MBC (62.5 µg/mL) values were observed against *S. epidermidis* (ATCC 12228) than for other donor samples. Finally, for donor 3 PRF_CLP samples, there is a difference in MIC values against *S. epidermidis* (clinical isolate); it is higher (125 μg/mL) than against *S. aureus* (ATCC 25923), *S. aureus* (clinical isolate), *S. epidermidis* (ATCC 12228)—62.5 μg/mL. A higher MBC value (250 μg/mL) is observed for clinical isolates of both bacteria. In turn, for the bacteria reference cultures, a lower MBC value (62.5 μg/mL) is observed against *S. epidermidis* than against *S. aureus* (125 μg/mL).

A U-shaped histogram is displayed in the test sections (see Figure 7). This is well observed in the negative control (PRF_CLP_broth), where the absorption capacity gradually decreases with increasing control dilution. The antibacterial data of all prepared PRF_CLP samples showed differences between the donors and the related MIC and MBC values of the samples (Figure 7). For PRF_CLP samples from donor 2 and donor 3 blood, all antibacterial data can be found in an additional file (Appendix A).

Comparing the results of PRF_CLP samples between *S. aureus* reference cultures and clinical isolates, it is observed that only for the donor 3 PRF_CLP samples require a higher CLP concentration (250 µg/mL) against the clinical isolate than against the reference culture (125 µg/mL) (Figure 7). Regarding MIC values, it was observed that only the donor 1 PRF_CLP samples against the clinical isolates required a higher CLP concentration (62.5 µg/mL) than against the reference culture (31.25 µg/mL).

From the results of PRF_CLP samples against the *S. epidermidis* reference culture and clinical isolate (Figure 7), we observed that there is a difference in MIC values for donor 2 PRF_CLP samples, with a higher CLP concentration to the clinical isolate (62.5 µg/mL) than to the reference cultures (31.25 µg/mL) being required to ensure antibacterial activity. MBC values required to provide antibacterial activity against both types of *S. epidermidis* bacteria differ from the *S. aureus* results described above. Looking at the results for donor 2 and donor 3 PRF_CLP samples, it was shown that a higher CLP concentration (125 µg/mL) was required against the *S. epidermidis* clinical isolate and a lower concentration against the *S. epidermidis* reference culture.

Differences in MIC and MBC values against a particular bacterial strain for all three donors are shown in Figure 6. Significantly, a higher MBC value—1000 µg/mL—was observed for CLP samples compared to all donor PRF_CLP samples against each bacterial strain. MBC value against *S. epidermidis* (ATCC 12228) decreased 16-fold for all donor PRF_CLP samples compared to the CLP samples, but decreased 8–16-fold against *S. epidermidis* (clinical isolate) and *S. aureus* (ATCC 25923). The efficacy of PRF_CLP samples against *S. aureus* (clinical isolate) is seen as a 4–16-fold reduction in the MBC value. Thus, indicating that the addition of the required CLP concentration to provide an antibacterial effect against the same bacteria varies greatly depending on the donor PRF. The average MIC and MBC values of all antibacterial data for all PRF_CLP samples can be found in an additional file (Appendix A).

### 2.4. Cell Viability

The obtained cell cytotoxicity results for PRF and PRF_CLP are shown in Figure 8. Fibroblasts are used for material testing because they have a wide range of functions in the human body, one of them being as part of connective tissue. As PRF has contact with fibroblasts in the body, it is important to test the biomaterial effect on them.

Precise results could be obtained after 24 h and 48 h. The reason for the vague results after 1 h, 2 h and 4 h is that PRF contains many cells, for example, leukocytes, monocytes, red blood cell platelets, neutrophils and lymphocytes [33], which affected cell staining (Figure 9). It should be noted that no difference was observed between PRF samples containing CLP and those not. The experiment had three controls—pure 10 mg/mL CLP solution, untreated cells (positive control) and cells treated with 5% DMSO (negative control).

Cell viability was above 70% for both PRF and PRF_CLP extracts and their dilutions were taken after 24 h and 48 h. According to ISO 10993-5:2009, a cytotoxicity effect is considered if the cell viability is decreased by more than 30% [34]. Dilutions did not show significant differences with extracts; however, PRF samples from different donors did have significant statistical differences (* *p* ˂ 0.05). With PRF_CLP samples, there is a small trend of cell viability increasing with dilution, but there is no trend visible with pure PRF samples, which indicates that dilution does not have a significant effect on cell viability.

## 3. Discussion

This study examined the ability of CLP to convert to clindamycin in the presence of PRF, to provide higher antibacterial activity than PRF and CLP alone. To date, no one has studied the hydrolysis of CLP in the blood without a specific chemical reaction, nor the ability of CLP to enhance the antibacterial properties of PRF. The structure, surface properties, antibacterial properties and drug release kinetics of PRF_CLP were tested.

As observed in the FTIR spectra, CLP interacts with PRF during the incubation for 7 days to provide partial hydrolysis and conversion to clindamycin. After seven days of incubation, a new bond formation and a phosphate group absorption maximum increase over time were observed, indicating structural changes that are likely to be a CLP switch to clindamycin. In addition, other impurities or degradation products than clindamycin can be formed during the degradation of clindamycin phosphate. Brown [19] mentions that in addition to free clindamycin, clindamycin 3-phosphate, clindamycin 4-phosphate and clindamycin 2-phosphate are formed during conversion. In contrast, Wang et al. [30] described the clindamycin phosphate degradation experiment, indicating that in addition to clindamycin, lincomycin-2-phosphate and a small amount of clindamycin B-2-phosphate are formed. Based on the obtained SEM results, we can conclude that the addition of CLP does not significantly affect the structure of PRF. Minor changes are observed during CLP degradation.

Release data suggest that the PRF_CLP sample can be used for one day local therapy, ensuring maximum CLP release within 1 h. Wang et al., 2020, combined clindamycin (2 µg/mL) with PRP, indicating that 90% of the administered dose was excreted within 10 min [35]. As we can see, our material is able to provide longer release kinetics. In the same way, the release time of the drug could be adjusted according to the required therapy by administering drug delivery systems.

The composition of the blood from each of the donors affects the antibacterial properties of the sample, specifically the amount of CLP required to achieve antibacterial activity. Comparing the MIC and MBC values of the PRF_CLP samples with pure CLP samples for all bacteria strains, a decrease in these values is observed with the addition of PRF to the CLP. The widespread increase in staphylococcal resistance to most antimicrobials, especially in resistant strains, points to the need for new effective treatments for staphylococcal infections [36]. Our antibacterial tests showed that the addition of PRF enhances the antibacterial activity of CLP not only against staphylococcal reference cultures but also against clinical isolates. It can be seen that against the clinical isolates of *S. aureus* and *S. epidermidis,* higher CLP concentrations are required in PRF_CLP samples to provide a lower MBC value compared to both bacteria reference cultures. Each donor has different blood properties (such as different white blood cell counts or vitamin D levels) that drastically affect the antibacterial effect and that is why we have such high error limits. All microbiological input data for all three donor PRF_CLP samples are shown in the supplement (Appendix A). The spread of the *S. aureus* strains that are resistant to certain antibiotics has been reported [37]. According to the Daum [38] and Naimi [39] studies, methicillin-resistant *S. aureus* (MRSA) isolates tend to be sensitive to clindamycin and are less likely to be resistant to antibiotics other than the β-lactam class. The same may be for the *S. epidermidis* clinical isolate. Studies from Schilcher et al. [40,41] and Kuriyama et al. [42] showed that the MIC of pure clindamycin in clinical isolates against MRSA can reach > 256 mg/L. Based on the review of the literature, studies have been performed to test the activity of CLP and clindamycin against dermally important microorganisms. The results showed that CLP had antimicrobial activity against the same organisms as clindamycin, with only a 3 to 44 times higher concentration dose [22]. Summarizing all the data, it can be seen that CLP with PRF is a better antibacterial material than pure CLP, and compared to the literature; we have obtained lower MIC values (ranging from 62.5 to 145.8 µg/mL depending on the bacterial strain) than required for clindamycin (>256 µg/mL) [40,41,42]. Depending on the bacterial strain, the concentration of the drug has to be adjusted.

To ensure that the obtained PRF_CLP matrices can be used for medical applications, in vitro cell viability tests were performed. The highest cell viability can be observed for 48 h extract and dilutions, where it increases above 100% for most of the samples. An increase in viability indicates that PRF increases cell proliferation [43]. PRF is known to be rich in transforming growth factor-β (TGF-β), platelet-derived growth factor (PDGF), vascular endothelial growth factor (VEGF) and epidermal growth factor (EGF) [44], which all have a significant role in new cell formation. Overall, CLP has a favorable effect on cell viability. By adding the antibiotic to PRF, the viability does not go below 80% in the extracts for the prepared time points. Navarro et al. [45] tested periodontal ligament (PDL) cell viability in PRF and concluded that PRF increases cell viability after PDL is exposed to PRF for 30 min, 1 h and 2 h. In this case, it can be noted that all the PRF ingredients have a favorable effect on PDL cells. The positive effect of PRF was also noticed in a Bucur et al. [46] study on the blood clot effect on fibroblast proliferation and migration. The samples were tested for 24 h and 48 h and in both cases, PRF positively affected cell viability; the same can be observed in our experiment. An interesting difference between the studies is that Bucur et al. filtered the testing solution before applying it to cells to remove blood cells. This should be taken into account for future experiments.

## 4. Materials and Methods

### 4.1. Materials

Clindamycin phosphate (CLP, Sigma Aldrich, St. Louis, MO 63103, USA), acetonitrile (≥99.9%, Sigma-Aldrich, St. Louis, MI, USA), phosphoric acid (H_3_PO_4_; C = 75% *w*/*w*, Latvijas ķīmija, Riga, Latvia), potassium dihydrogen phosphate (KH_2_PO_4_, Sigma Aldrich, ≥99%), methanol (≥99.9%, Sigma-Aldrich, St. Louis, MI, USA), Dulbecco’s Modified Eagle’s Medium (DMEM, Sigma Aldrich, St. Louis, MO, USA), bovine calf serum (CS, Sigma-Aldrich, St. Louis, MO, USA), Penicillin/Streptomycin (P/S, Sigma-Aldrich, St. Louis, MO, USA), dimethylsulfoxide (DMSO, Sigma-Aldrich, St. Louis, MO, USA), neutral red (NR, Sigma Aldrich, St. Louis, MO, USA), Phosphate Buffer Saline (PBS, Sigma-Aldrich, St. Louis, MO, USA), acetic acid (Sigma-Aldrich, St. Louis, MO, USA), ethanol (96%, Latvian Chemistry, Riga, Latvia).

### 4.2. Blood Collection and Platelet-Rich Fibrin Production

Blood of 3 healthy volunteers with vitamin D levels > 30 ng/mL was collected in 13 mL i-PRF+ tubes (PROCESS FOR PRF, 06000 Nice, France) and immediately placed in a centrifuge (“PRF DUO Quattro”). PRF was obtained by centrifugation at 700 rpm for 5 min (for women) or 6 min (for men). After the centrifugation, the upper layer of liquid PRF (1 mL) from one donor of each tube was transferred into a 50 mL tube, and mixed together for further use.

An amount of 0.5 mL of liquid PRF was used to obtain one PRF sample. To prepare PRF samples with CLP (PRF_CLP), 0.5 mL PRF was added to pre-weighed 0.5 mg CLP with an automatic pipette and mixed well with a spatula. Samples for FTIR and SEM analysis were prepared by incubation (Environmental Shaker-Incubator ES-20, Biosan, Riga, Latvia) at 37 °C for 1, 3 and 7 days and then lyophilized for 72 h. For drug release and cell experiments, coagulated PRF and PRF_CLP samples were used.

Written consent from all of the volunteers for use of their samples in the research studies was obtained. All donors were free of any infectious disease and had no abnormal nicotine or alcohol use. None of the subjects used any anticoagulant drugs. Permission No. 6-2/10/53 of the Research Ethics Committee of Riga Stradins University was obtained for the study.

### 4.3. Characterization of Prepared Samples

#### 4.3.1. Chemical Structure

The lyophilized PRF and PRF with CLP (PRF_CLP) samples after 1, 3 and 7 days of incubation were investigated with Fourier-transform infrared spectroscopy (FTIR) attenuated total reflection (ATR) method, to identify functional groups in PRF matrix. ATR spectroscopy spectra were taken with Thermo Fisher Scientific Nicolet iS5 with a diamond crystal. Spectra were recorded from 500 to 4000 cm^−1^ with 64 scans and with a resolution of 4 cm^−1^, optical velocity 0.4747, and aperture 100%.

#### 4.3.2. Morphology

Scanning electron microscope Tescan Mira/LMU (Tescan, Brno, Czech Republic) was used to visualize the microstructure and morphology of obtained PRF and PRF_CLP samples. Prior to examination, samples were fixed to aluminum pin stubs with conductive carbon tape and sputter coated with thin layer of gold at 25 mA for 3 min using Emitech K550X (Quorum Technologies, Ashford, Kent, UK). Secondary electrons created at 5 kV were used.

#### 4.3.3. Evaluation of CLP Kinetics

Evaluation of CLP release kinetics was analyzed using ultra-performance liquid chromatography. The chromatographic method was adapted based on other studies [47,48]. A chromatograph ”Waters Acquity UPLC H-class” with a UV/VIS detector ”Waters Acquity TUV” at 195 nm and column ”Waters Acquity UPLC BEH C18, 1.7 µm, 2.1 × 150 mm” was used for data acquisition. The mobile phase consisted of 0.02M KH_2_PO_4_ buffer (PH = 2.5 ± 0.02): acetonitrile in ratio 79:21, respectively, and at a flow rate of 0.3 mL/min. The total analysis time for one sample was 7 min. During the analysis, the column temperature was maintained at 40 °C ± 5 °C and the sample temperature at 10 °C ± 5 °C. The limit of quantification and the limit of detection for the developed method were found to be 1.157 µg/mL and 0.382 µg/mL.

Samples for CLP release studies were immersed in 20 mL of DMEM and placed in an incubator at 37 °C ± 5 °C. At the first 2 time points (15 min and 30 min), the solution was completely removed, and at the other time points (1 h, 1.5 h, 2 h, 4 h, 6 h, 17 h, 24 h), 2 mL aliquots of the solution were used. Finally, 2 mL of DMEM was returned after each sample to ensure a constant volume during the release experiment.

### 4.4. Preparation of Bacterial Suspension

Four bacterial strains were used in the study, reference culture of *S. aureus* (ATCC 25923) and *S. epidermidis* (ATCC 12228), and clinical isolates of *S. aureus* and *S. epidermidis*, which previously were isolated from the pure sample and identified with VITEK2 system (bioMérieux, Marcy l’Etoile, France). Before the antibacterial tests, bacterial susceptibility against clindamycin was tested with the disc diffusion method. All bacterial suspensions were prepared according to EUCAST (European Committee on Antimicrobial Susceptibility Testing) standards in optic density of 0.5 according to McFarland standard with optic densitometer (Biosan, Riga, Latvia). All bacterial strains showed sensitivity against clindamycin (2 µg) discs (Liofilchem S.r.l., Roseto degli Abruzzi, Italy).

### 4.5. Determination of Antibacterial Properties

The antibacterial tests were investigated with EUCAST (European Committee on Antimicrobial Susceptibility Testing) standard laboratory antibacterial susceptibility testing method—broth microdilution (Figure 10) [49,50].

#### 4.5.1. Determination of Minimal Inhibitory Concentration

Three different test sample solutions were used: pure PRF, PRF_CLP and pure CLP. Samples with PRF were diluted 1:5, accordingly 2 mL PRF and 8 mL Mueller–Hinton broth (MHB) (Oxoid, UK) to obtain 2 mg/mL stock solution of CLP. A 96-well plate (SARSTEDT, Nümbrecht, Germany) was used in the quantitative assay. Twofold serial dilutions of the pure CLP and PRF_CLP stock solutions (ranging between 2000 and 7.8125 µg/mL) were performed in a 100 μL volume. Each well was seeded with 100 μL of bacterial suspension (10^6^ CFU/mL, 0.5 McFarland density), where 200 μL of pure bacterial suspension (10^6^ CFU/mL) served as positive control while pure sterile MHB served as negative controls. To detect the MIC and MBC values we used PRF_CLP controls in broth with and without bacteria. After 2-fold dilution, instead of adding bacterial suspension, sterile MHB was added. Then, 96-well plates were incubated in a thermostat (Memmert GmbH, Schwabach, Germany) for 18 h at 37 °C. MIC values were considered as the lowest concentration of the tested solution that inhibits bacterial growth in microdilution wells as visually detected. After incubation, the values of absorbance were measured with microplate reader at 570 nm (Tecan Infinite F50, Männedorf, Switzerland).

#### 4.5.2. Determination of Minimal Bactericidal Concentration

The lowest concentration at which bacterial growth was completely inhibited by the additional culture method on non-selective media was taken as the MBC value. To determine MBC, extra cultivation of 10 µL samples from the wells (prepared according to the methodology specified in Section 4.5.1) were inoculated on non-selective agar plates (Oxoid, UK); one sample from the well above the MIC value and all remaining below MIC value (MIC values based on data from methodology 4.5.1 were used). Agar plates were incubated in a thermostat (Memmert GmbH, Schwabach, Germany) for 18 h at 37 °C.

### 4.6. Cell Viability Experiments

PRF with and without CLP was tested on 3T3 mouse fibroblasts. Overall PRFs from 3 different donors were tested.

Prior to cell viability tests, 5000 cells were seeded in a 96-well plate in 200 µL of full cell medium. To prevent the plates from drying out, PBS was added to the outer wells. After seeding the cells, the plates were incubated overnight (37 °C, 5%) (New Brunswick™ S41i CO_2_ Incubator Shaker, Eppendorf, Hamburg, Germany).

The following day each PRF sample with and without CLP was submerged in 2 mL of full cell medium. The medium consisted of 89% DMEM, 10% CS and 1% P/S. After 1, 2, 4, 24 and 48 h, all the solution was removed from the testing sample and replaced with a fresh 2 mL cell medium. Extract and 2 types of dilutions—1:10 and 1:100—were directly put onto the cells. Before adding the analyzing solution to the cells, the old medium was removed. The experiment had two types of controls. The positive control consisted of untreated cells with medium; on the other hand, for the negative control, 5% DMSO solution in cell medium was applied to cells to analyze their viability. Each treatment had 6 replicates.

To analyze the PRF extract and its dilutions effect on cell viability, Natural Red (NR) test was used. The tests included PBS, NR and solubility solution (1% acetic acid, 50% ethanol, 49% water).

After 24 h of each time point, the testing solutions were discarded and cells were washed with 200 μL PBS solution. Subsequently, cells were treated with 150 μL NR solution, after which plates were left to incubate for 2 h. Afterward, the solution with dye was taken off, and cells were washed again with 250 μL PBS solution. Finally, cells were solubilized, which was done with a 150 μL solubilization solution. Then, a 540 nm wavelength was used to measure optical density with a microplate reader (Tecan Infinite M Nano, Switzerland). Every plate was analyzed appropriately with the method just described.

### 4.7. Statistical Evaluation

All results are expressed as the mean ± standard deviation (SD) of at least three independent samples. The reliability of the results was assessed using the unpaired Student’s t-test with a significance level of *p* < 0.05. One—and two-way analysis of variance (ANOVA) was performed to assess the differences between the results.

## 5. Conclusions

The results of the present study show the structure, surface properties, antibacterial properties, drug release kinetics, and cell viability of the PRF_CLP samples. Burst release (80% of CLP after 1 h) was observed for the PRF_CLP samples; thus, the development of more advanced drug delivery systems could be an area for future research. The antibacterial effect of CLP was affected by the addition of PRF, thus providing a reduction in MIC and MBC concentrations compared to pure CLP and pure PRF samples. Cell viability for the PRF_CLP samples increased indicating the ability of PRF to alter cell proliferation. Structural studies have also shown that clindamycin phosphate is converted to clindamycin within the PRF matrix at 37 °C.

This study proves that the presence of PRF in the resulting PRF_CLP samples improves the antibacterial efficacy and may be suitable for medical applications. The results are the first step in finding alternative solutions that can enhance the antibacterial properties of CLP to prevent postoperative infections and could lead to a new method to be developed, which may increase the efficiency of drug delivery and activity. Further clinical trials with larger patient groups are needed to introduce this method for reducing the risk of post-operative infections.

## Figures and Tables

**Figure 1 ijms-23-07407-f001:**
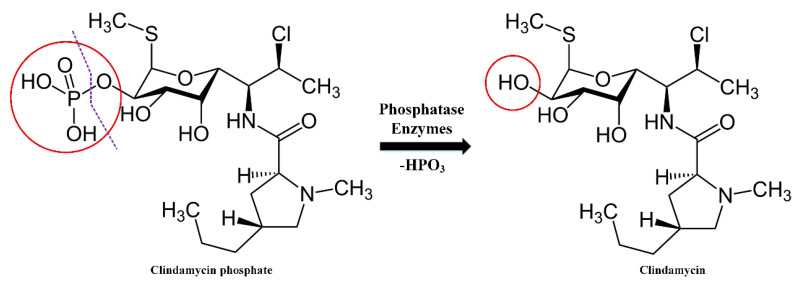
Clindamycin phosphate hydrolysis mechanism.

**Figure 2 ijms-23-07407-f002:**
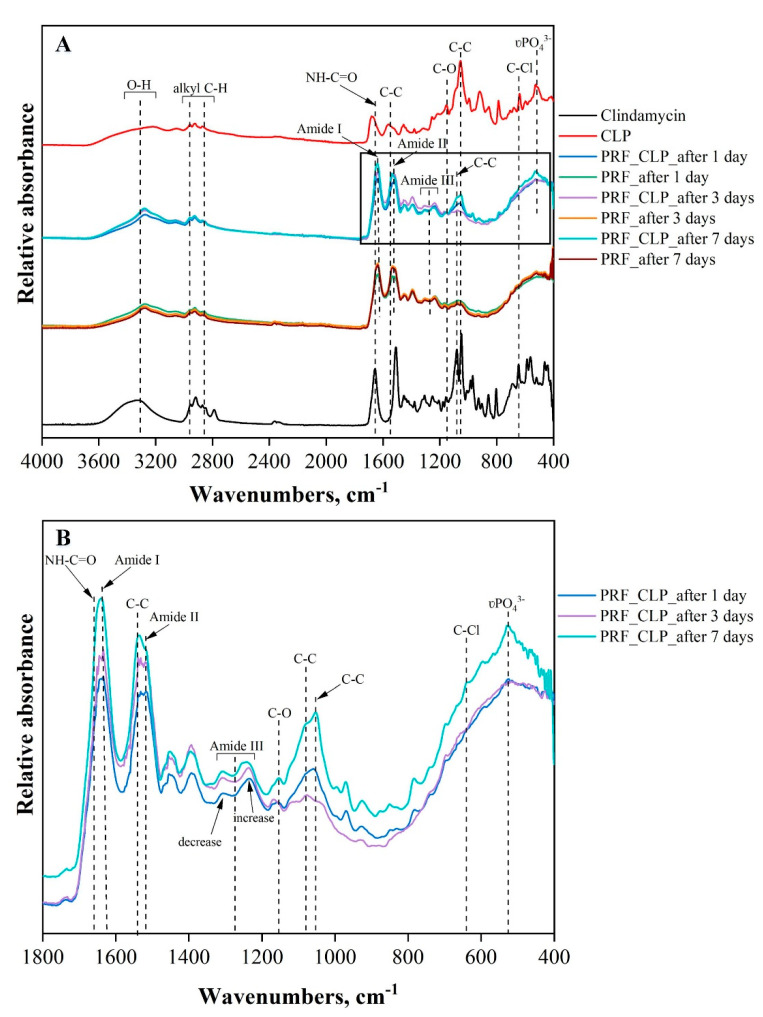
FTIR spectrum: (**A**) Full spectrum of absorption peaks PRF, PRF_CLP samples, CLP and clindamycin; (**B**) FTIR spectrum of PRF_CLP samples at incubation time points.

**Figure 3 ijms-23-07407-f003:**
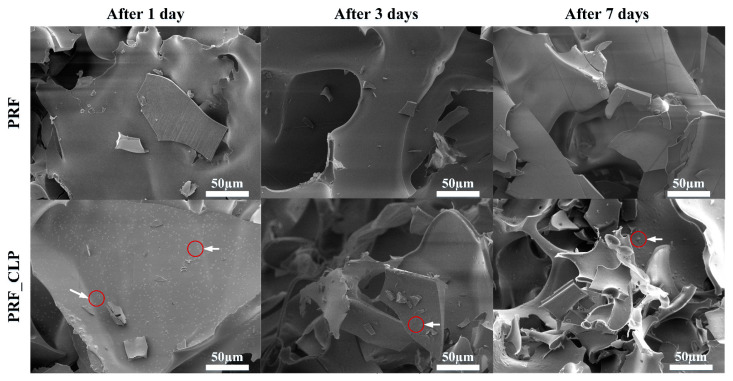
SEM pictures of PRF and PRF_CLP matrix surface; red circles with white arrows indicate the existence of NaCl in the PRF samples.

**Figure 4 ijms-23-07407-f004:**
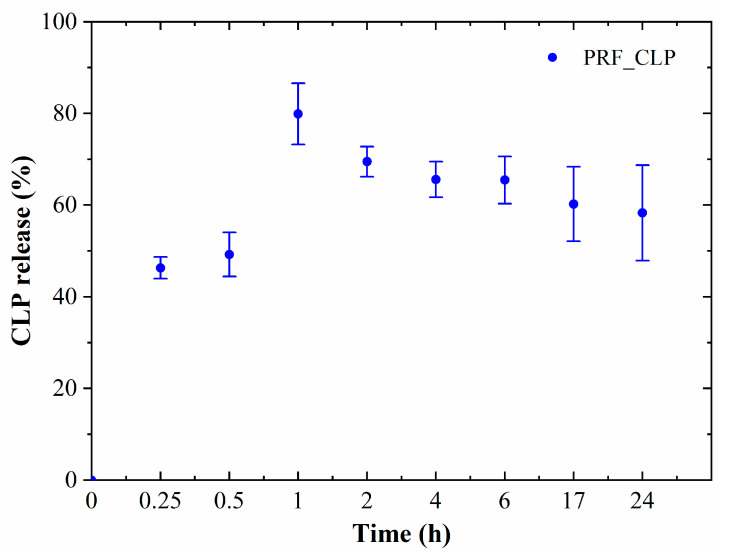
CLP release from PRF matrices in DMEM; average of 3 donor release data.

**Figure 5 ijms-23-07407-f005:**
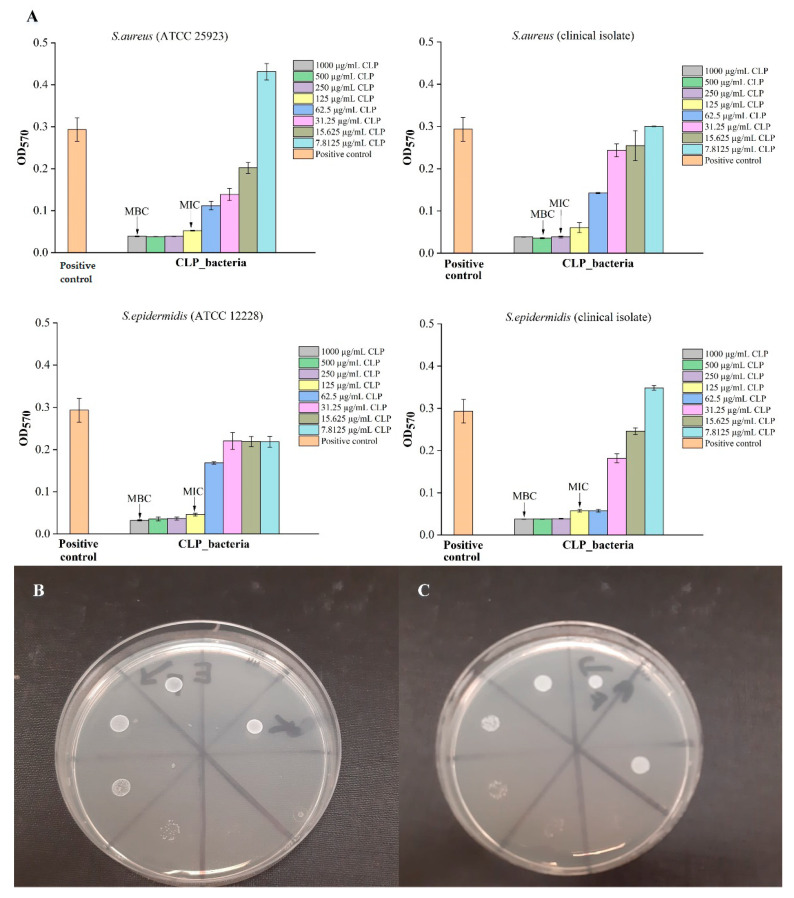
Antibacterial properties of different CLP solutions at various concentrations: (**A**) detected MIC and MBC concentrations for 4 bacteria (*S. aureus* (ATCC 25923), *S. epidermidis* (ATCC 12228), *S. aureus* (clinical isolate), *S. epidermidis* (clinical isolate); (**B**) MBC test for *S. aureus* (clinical isolate); (**C**) MBC test for *S. aureus* (ATCC 25923). The diameter of the Petri dishes is 8.5 cm.

**Figure 6 ijms-23-07407-f006:**
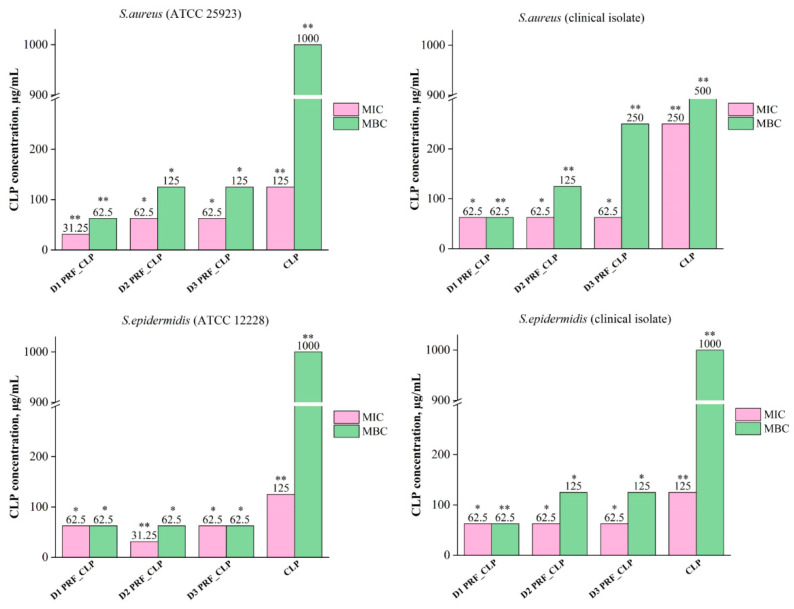
MIC and MBC value differences between CLP and PRF_CLP samples against four bacteria stains (*S. aureus* (ATCC 25923), *S. epidermidis* (ATCC 12228), *S. aureus* (clinical isolate) and *S. epidermidis* (clinical isolate) for all three donors. Samples prepared from donor 1 blood (D1 PRF_CLP); samples prepared from donor 2 (D2 PRF_CLP); samples prepared from donor 3 (D3 PRF_CLP). * *p* > 0.05; ** *p* <0.05.

**Figure 7 ijms-23-07407-f007:**
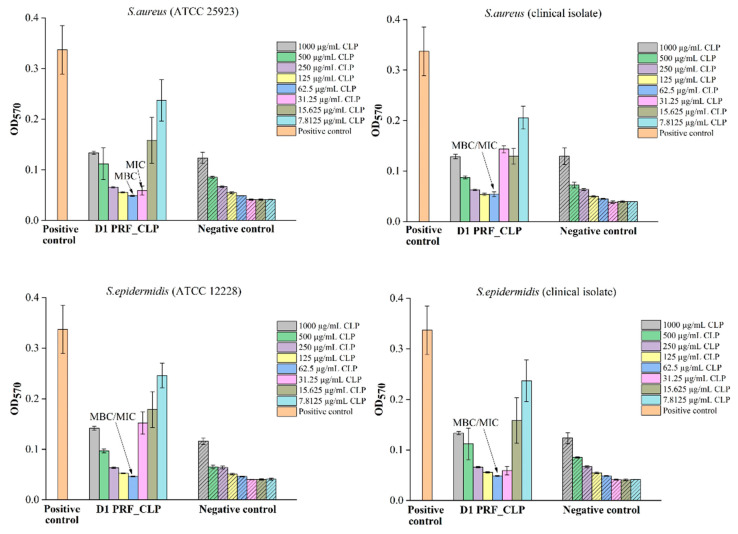
Antibacterial properties of PRF_CLP samples at various concentrations of CLP solution for 4 bacteria strains (*S. aureus* (ATCC 25923), *S. epidermidis* (ATCC 12228), *S. aureus* (clinical isolate) and *S. epidermidis* (clinical isolate) for PRF_CLP samples prepared from donor 1 blood. Pure bacterial suspension (10^6^ CFU/mL) as a positive control and pure sterile Mueller–Hinton broth as a negative control were used.

**Figure 8 ijms-23-07407-f008:**
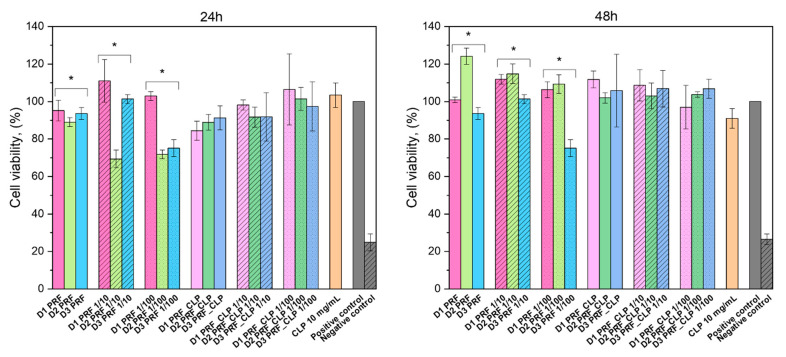
Cytotoxicity of PRF and PRF-CLP extracts and dilutions (significant statistical difference (* *p*˂ 0.05)).

**Figure 9 ijms-23-07407-f009:**
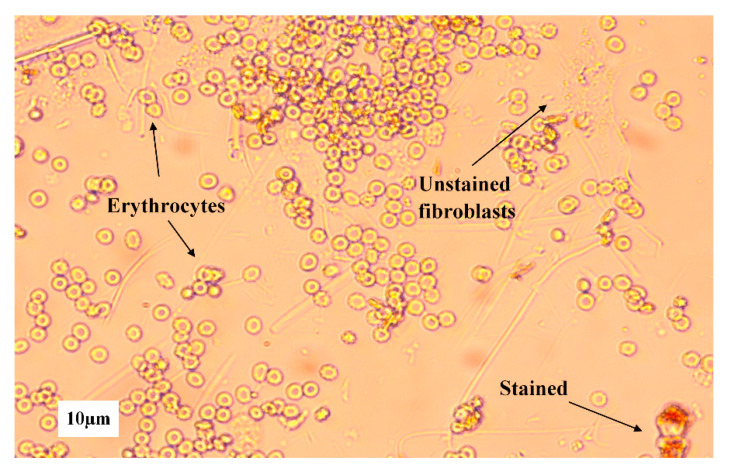
Different blood cells on 3T3 fibroblast cells from D2 PRF sample extract taken after 1 h.

**Figure 10 ijms-23-07407-f010:**
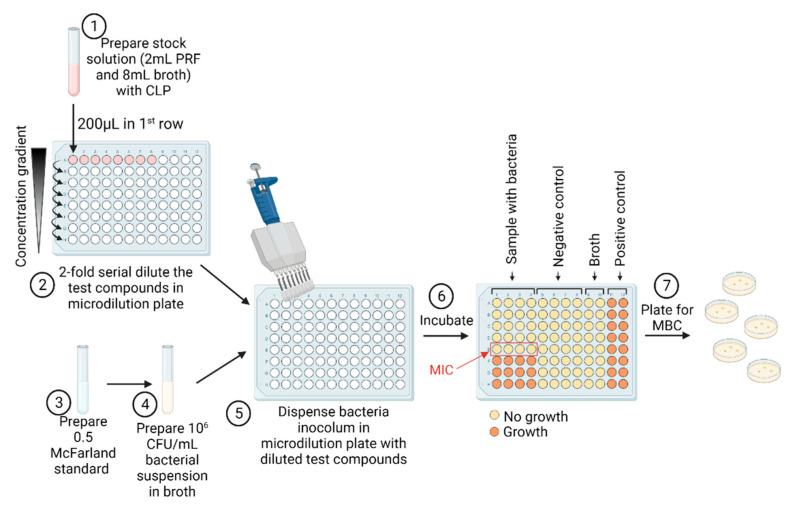
The MIC/MBC assay of CLP, PRF and PRF_CLP samples. Figure created with Biorender.com.

## Data Availability

The data presented in this study are available on request from the corresponding author.

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
