# Peer review of "Injectable Platelet-Rich Fibrin as a Drug Carrier Increases the Antibacterial Susceptibility of Antibiotic—Clindamycin Phosphate"

_ijms, 2022, doi:10.3390/ijms23137407_

Round 1

Reviewer 1 Report

The manuscript by Egle et al. is original, well designed and conducted, with relevant contribution to the understanding of the antibacterial effects of clindamycin phosphate (CLP), in combination with platelet-rich fibrin (PRF) against S. aureus and S. epidermidis. Furthermore, clindamycin phosphate in combination with platelet-rich fibrin showed a low level of toxicity after 24 h and 48 h The results constitute an important step in identifying alternative solutions to improve the antibacterial properties of CLP, in order to prevent postoperative infections in humans.

I support its publication after appropriate minor modifications as outlined below.

1.     Rephrasing of the abstract, highlighting the key findings obtained in the present study, omitting the unimportant aspects and point out the limitations of this study and future perspectives.

2.     Results section: please quote some articles (lines 104-107). The authors need to explain why they chose only 3 donors? Does the low number of donors statistically cover the research results?

Subchapters should be italicized.

3.     Discussion section: authors need to check that each reference cited in the text has a number in the reference list, and also correct minor errors in the text when quoting the authors.

4.     Materials and Methods section: the authors need to provide references for the testing protocols. Please correct cell density (line 436).

5.     The authors should rephrase the conclusions, including practical applicability, as well as the future perspectives.

6.     Some bibliographic references are incomplete. Please check them carefully and complete the missing data, according to the journal requirements.

7.     Lines 53, 54, 61, 62, 90, 94, 418, 419, 420, 530, 531, 536, 539, 541, 544, 550, 580, 607, 609, 611, 613, 614, 617: the name of the microorganism species are not italicized. Please be carefully with this basic concern throughout the manuscript!

Author Response

Thank you for the review, your comments and suggestions are valuable to us and we have improved the manuscript.

Reviewer 2 Report

The topic of the study is very interesting and worth of exploration. The results contained in the manuscript contribute to an interesting field of treatment improvement using platelet-rich fibrin as a carrier system of drugs/prodrugs. This detailed study could allow to develop a new method which may increase efficiency of drug delivery and activity.

Strength of the study is novel approach to drug delivery with applicable potential, advanced and deep insight into complex structure with use appropriate methodology. I have no big objections to the quality of the manuscript: methods are appropriate, all parts of the manuscript (introduction, results and discussion) are quite decent.

However, I will address some points to the study which could help to improve the manuscript:

1. As a microbiologist I object to calculate mean values for MIC and MBC from repeated samples. The recommended description of these values is presentation of the highest value found during testing or range of values. So, it is appropriate to use concentration values instead of mean values which are  artificial numbers in this context.

2. All bacterial species names should be written in italics.

3. In the Abstract abbreviations should be explained (FTIR, SEM, UPLC).

4. What was the reason for the use of EUCAST recommendation for disk diffusion method and CLSI recommendation for microdilution method? In Europe EUCAST is the operative standard.

5.  In the Methods (p. 13, line 446) there is mistake in MIC definition – first MBC should be changed to MIC.

6. In the Discussion (p. 11, line 320), I do not know the word “antibacteriality”. It should be corrected.

7. In the Discussion (p.11, line 322), the term “S. aureus colonies” should be changed to “S. aureus strains”.

8. The statement in the Discussion (p. 11, line 336) needs a reference. In my opinion it is rather awkward statement because there are also many S. epidermidis strains which are more resistant than S. aureus strains. I do not know, what the authors meant. I suppose that the authors thought about targeted antibiotic therapy.

Author Response

(The authors gave the same response as above.)
